# The impact of long-term conditions on the progression of frailty

Ali Alattas[1]*, Silviya Nikolova[2], Farag Shuweihdi[1], Kate Best[1], Robert West[1]

1 Leeds Institute of Health Sciences, School of Medicine, University of Leeds, Leeds, United Kingdom,
2 EMEA Real World Methods & Evidence Generation, IQVIA, London, United Kingdom

* mmaalat@leeds.ac.uk

**Data Availability Statement:** Researchers can access the ELSA dataset for free from the UK Data Service. You can assess the data here beta. ukdataservice.ac.uk/datacatalogue/series/series?id=200011.

## Abstract

### Objective

To explore longitudinally the impact of multiple long-term conditions (LTCs) on frailty progression separately for males and females.

### Methods

A functional frailty measure (FFM) was used to examine putative determinants of frailty progression among participants aged 65 to 90 in the English Longitudinal Study of Ageing (ELSA), across nine waves (18 years) of data collection. A multilevel growth model was fitted to measure the FFM progression over 18 years, grouped by LTC categories (zero, one, two and more).

### Results

There were 2396 male participants at wave 1, of whom 742 (31.0%) had 1 LTC and 1147 (47.9%) had ≥2 LTCs. There were 2965 females at wave 1 of whom 881 (29.7%) had one LTC and 1584 (53.4%) had ≥2 LTCs. The FFM increased 4% each 10 years for the male participants with no LTCs, while it increased 6% per decade in females. The FFM increased with the number of LTCs, for males and females. The acceleration of FMM increases for males with one long-term health condition or more; however in females the acceleration of FMM increases when they have two LTCs or more.

### Conclusion

Frailty progression accelerates in males with one LTCs and females with two LTCs or more. Health providers should be aware of planning a suitable intervention once the elderly have two or more health conditions.

**Funding:** This work was funded by the Cultural Bureau, Embassy of Saudi Arabia, London. The funders had no role in study design, data collection and analysis, decision to publish, or preparation of the manuscript.

**Competing interests:** The authors have declared that no competing interests exist.

## Introduction

Population ageing leads to increased demand for health and social care and associated cost pressures [1]. By 2028, 25% of England's population will be aged 65 and over [2], with 8% classified as frail [3]. Frailty increases with age and is associated with higher healthcare utilization [4], and its determinants are key for effective healthcare services provision. Studies have identified protective (e.g. higher wealth, increased social support) and harmful (e.g. lower wealth, educational achievement, presence of long-term conditions, being female) factors associated with frailty progression [5]. However, there is a lack of evidence on the impact of multiple long-term conditions (LTCs) longitudinally as a separate determinant of frailty progression [5]. LTCs are defined as "A long term condition is one that cannot currently be cured but can be controlled with the use of medication and/or other therapies" [6]. Sanders, Boudreau [7] studied the impact of diabetes on frailty development, and Thompson, Theou [8] reported that two or more LTCs contributed to increasing frailty in older people.

This study aimed to explore longitudinally the impact of multiple LTCs on frailty progression separately for males and females due to behavioural, social, and biological differences [9].

## Methods

### Data and analytical sample

The English Longitudinal Study of Ageing (ELSA) was used in this study. ELSA is a longitudinal study taken from private households of people in England aged over 50 [2]. It attempts to reflect the population profile of older people living in England, so it collects information on three aspects of ageing: health, social participation and wellbeing, and finances. ELSA currently features nine waves of data collected over 18 years [10]. We employed unit weights for each participant contribution since no appropriate weighting was provided by the ELSA team to ensure representation of the English population for all participants, aged 65 and above, across all nine waves.

Fig 1 shows the flowchart for the analytical sample. In the first stage, we determined the eligible participants, and at stage two, we explained how we handled missing data. The data analysis was conducted for females and males separately and so numbers for each sex are included in the flowchart. The participants under 65 years old were excluded since Searle, Mitnitski [11] reported that the age of 65 is considered a threshold for the start of an exponential relationship between frailty and age. In practice, higher levels of frailty are few among those under 65, so our focus is on those participants of ELSA for whom frailty progression is an important issue.

### Outcomes

**Frailty measure.**   The cumulative deficit model is one way to measure frailty in older people [12]. This model uses a range of deficits that cover five dimensions: symptoms, signs, abnormalities, diseases status and disability [12]. It is preferred since the model predicts mortality more accurately [13], and is more sensitive to a person's health status [14].

Marshall, Nazroo [15] have proposed a frailty index using the cumulative deficit model that is based on 62 deficits available in the ELSA dataset. Two deficits were removed from the memory test domain, "Prompt given for prospective memory test" and "number of animals mentioned" as they were not collected in all waves. Table 3 in the S1 Text provides details of the deficits. The deficits were scored as one if present or zero if absent. Items with a range of values are converted to values between 0 and 1 to indicate the severity of the defect.

Since the aim of this work is to study the impact of multiple LTCs on frailty and frailty progression, we drop the 16 long-term conditions from the remaining 60 items of the ELSA frailty

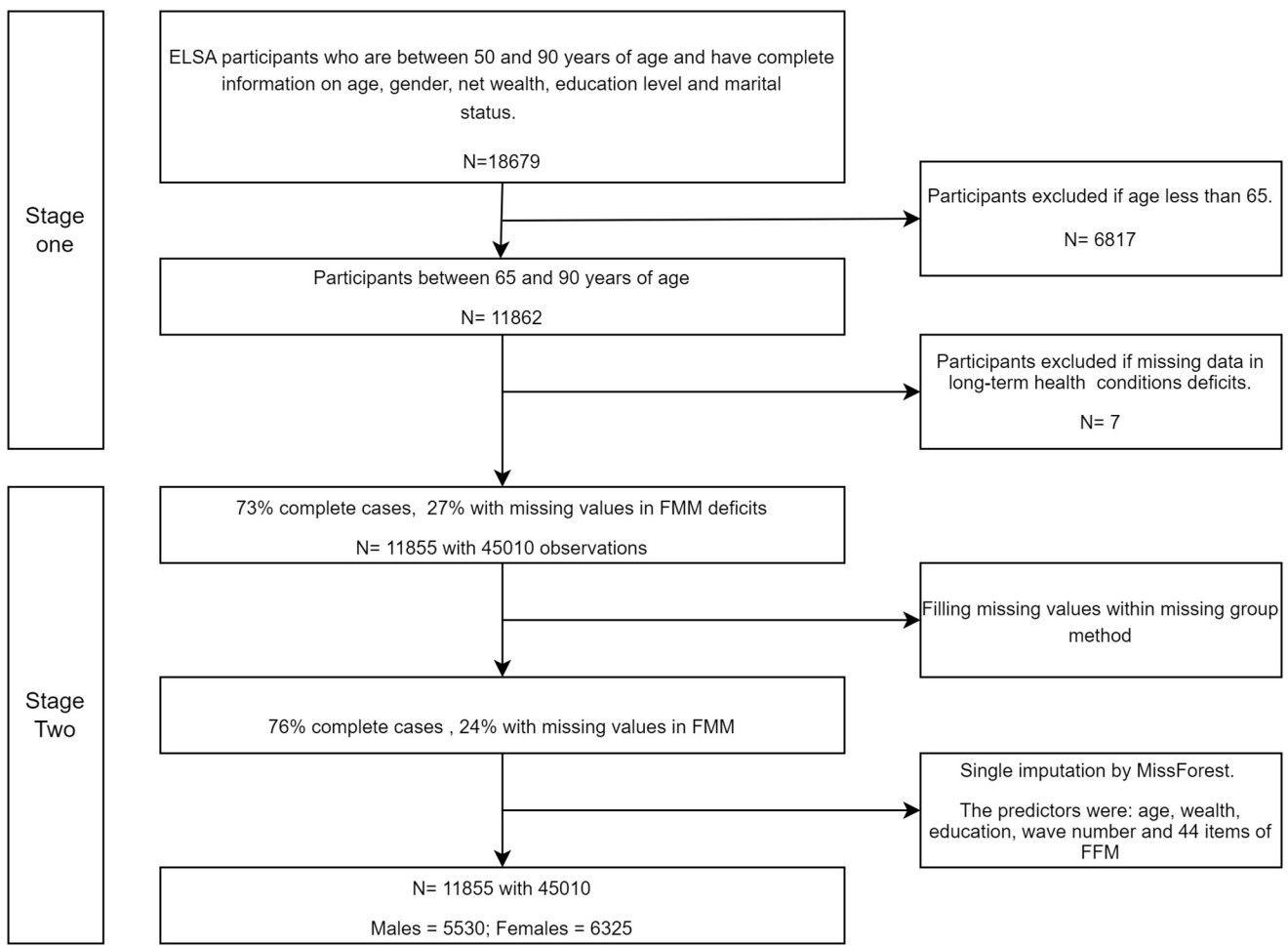

**Fig 1. The flowchart for the analytical sample + handled missing data.**

index to avoid mathematical coupling. The remaining 44 deficits were used to construct a Functional Frailty Measure (FFM). A similar approach has been taken by who extracted the morbidities from the frailty index to investigate the impact of comorbidity count on frailty development. Wade, Marshall [16] omitted pain and depression deficits from the frailty index to study their impact on frailty development.

We assessed the validity of the FFM by examining the relationship between all-cause mortality and FFM. Results suggest that FFM is a strong predictor of mortality (see Table 4 in the S1 Text). The FFM also satisfies the requirement of Searle, Mitnitski [11] in having more than 30 deficits. As a consequence of these aspects, FFM is a suitable measure for frailty.

**Determinants.** The list of 16 LTCs within the 60-item Frailty Index includes hypertension, angina, heart attack, congestive heart failure, abnormal heart rhythm, diabetes, stroke, lung disease, asthma, arthritis, osteoporosis, cancer, Parkinson's, psychiatrist, Alzheimer's, and dementia. Our primary interest is in the impact of multimorbidity, defined as two or more LTCs based on the National Institute for Health and Care Excellence [17]. Thus, the total count of LTCs was classified into three categories; none, one, and 'two or more' LTCs. In additions, three secondary determinants were selected: age, education level and net wealth with the removal of non-pension wealth. These four determinants were chosen because they were

commonly used in comparative studies [15, 18] and had few missing values across the nine waves of ELSA (see Table 5 in the S1 Text). Education was aggregated into three categories. We included NVQ 4/NVQ 5/Degree or higher degree in a high degree category and had no qualifications as a separate category named low education. The remaining educational qualifications were banded as a separate group of average educational attainment and included NVQ 3/GCE Advanced Level or NVQ 2/GCE Intermediate Level or NVQ 1/CSE Foundation level or another grade or foreign/other. The net wealth includes the sum of savings, investments, physical wealth and housing after financial debt is subtracted. It is adjusted for inflation using the Consumer Price Inflation Index (CPIH which includes housing costs). The base year was 2015 (see Table 6 in the S1 Text). The net wealth is classified initially by quintiles: richest, rich, average, poor, and poorest, and the upper two classes and lower two classes were combined to produce three categories: rich, average and poor.

## Statistical approach

**Handling missing data.** Failure to deal with missing data in longitudinal studies can lead to biased and inefficient statistical analyses [19]. We handled the missing data in two ways. First, the missing values were filled in using information from within the missing group. For example, if an individual has a missing value between two reported waves for a deficit, the missing value for a deficit is replaced with the same value. Second, for any remaining missing values, the MissForest algorithm was applied [20]. Although it is a single imputation method, it has the benefit of accommodating the nonlinearities and interactions for the predictors and is comparable to multiple imputation methods [21].

A participant who reports a LTC at any wave is assumed to have the condition at subsequent waves. Due to the low rate of missing values in the secondary determinants, which are age, net wealth and education level, these observations were deleted (see Table 5 in the S1 Text).

The MissForest algorithm uses two-thirds of non-missing observations to build a prediction model and test the model by predicting one-third of non-missing observations or what is well-known out-of-bag observations. The error percentage for the MissForest was used as a norm to evaluate the imputation.

The total rate of missing data on the FMM deficits was 27%. The first approach, which is filling the missing data within group value, reduced the rate of missing data to 24%. The error percentage for the MissForest prediction was 12%.

**Regression splines.** It is an option to deal with the non-linear relationship between the dependent and the number of independent predictors. Here the regression spline was applied as an initial investigation of the relationship between FFM and age grouped by 1) gender and 2) LTCs so that it guides the selection of the multilevel growth model which can adjust for other covariates.

The general idea for the regression splines method is dividing the range of a dependent variable by specific points called "knots", then using a piecewise cubic function to estimate the curve for each part. In general, it is preferred to add more knots in a region of high curvature and use fewer knots in flat regions. We applied this method in our analysis by using *bs* function in R which uses B-splines [22].

**Multilevel growth model.** A multilevel growth curve model was fitted to predict changes in the FFM over the 18 years covering the 9 waves accounting for several determinants. Multilevel models consider the non-independence of an individual's scores on the frailty index over time [23]. FFM was treated as time-varying continuous variables at nine-time points, and due to a nonlinear relationship between frailty and age, the quadratic term of age was added to the

full model. The entire model of FFM included: age, age$^2$, education level, net wealth, LTC categories and two-way interaction between age and LTC categories were added as fixed effects as well as age and age$^2$ as random effects.

The upper limit of age was 90, and it was included as a continuous covariate centred around 70 and then divided by 10 to ease interpretation. All three determinants selected, which are age, education and net wealth, were time-varying.

Implementing multilevel models in longitudinal data might be more complicated because an individual's current measure may often be correlated to prior measures [24]. Using the autocorrelation function (ACF), we observed autocorrelated errors within individuals (see Fig 3 in the S1 Text). Consequentially, we used robust standard errors (RSE) [25]. Robust standard errors give a growth model that is robust against autocorrelation and heteroskedasticity.

Restricted maximum likelihood (REML) estimates were used. We turn to the standard maximum likelihood method if we only want to compare nested models with different fixed effects [26]. To find the better model, Akaike's information criterion (AIC) was used. It is calculated as (AIC = -2(log-likelihood) + 2$K$), where likelihood is a measure of model fit, and $K$ is the number of model parameters. Each model will be ranked by the AIC from best to worst and then the higher ranked model will be selected [22]. In addition, the ANOVA function in R was utilized to see whether there was a significant difference between compared models. The package *nlme* (V3.1–155) [27] for R (4.2.2) was used to conduct the analysis. We conducted a sensitivity analysis using a multilevel growth model after excluding all observations with missing data.

## Results

### Sample characteristics

Tables 1 and 2 show the summary statistics for all determinants and FFM scores across the nine waves for males and females. The sample size across the nine waves for males ranges between 1856 and 2489 participants and between 2333 and 2489 for females. Across the nine waves, there are more female participants than males.

In males and females, due to the refreshment samples in waves 3,4,6,7, and 9, the average age over time is between 73 and 74 years old. Also, high- or medium-education participants

**Table 1. Summary statistics for five determinants and FMM for males across the nine waves.**

|  | Wave | 1 | 2 | 3 | 4 | 5 | 6 | 7 | 8 | 9 |
|---|---|---|---|---|---|---|---|---|---|---|
|  | N | 2396 | 2039 | 1856 | 2223 | 2326 | 2489 | 2405 | 2383 | 2353 |
| Age (M, SD) |  | 73.40 (6.22) | 73.51 (6.18) | 73.90 (6.28) | 73.28 (6.05) | 73.50 (6.25) | 73.35 (6.43) | 73.54 (6.36) | 73.59 (6.31) | 73.95 (6.38) |
| Education (n, %) | High | 467 (19.5) | 474 (23.2) | 571 (30.8) | 753 (33.9) | 850 (36.5) | 902 (36.2) | 889 (37.0) | 890 (37.3) | 1090 (46.3) |
|  | Med or foreign | 794 (33.1) | 722 (35.4) | 670 (36.1) | 790 (35.5) | 841 (36.2) | 926 (37.2) | 962 (40.0) | 974 (40.9) | 865 (36.8) |
|  | low | 1135 (47.4) | 843 (41.3) | 615 (33.1) | 680 (30.6) | 635 (27.3) | 661 (26.6) | 554 (23.0) | 519 (21.8) | 398 (16.9) |
| Net wealth (n, %) | Rich | 921 (38.4) | 794 (38.9) | 750 (40.4) | 924 (41.6) | 987 (42.4) | 1078 (43.3) | 1071 (44.5) | 1038 (43.6) | 1097 (46.6) |
|  | Average | 473 (19.7) | 426 (20.9) | 399 (21.5) | 456 (20.5) | 488 (21.0) | 514 (20.7) | 515 (21.4) | 495 (20.8) | 480 (20.4) |
|  | Poor | 1002 (41.8) | 819 (40.2) | 707 (38.1) | 843 (37.9) | 851 (36.6) | 897 (36.0) | 819 (34.1) | 850 (35.7) | 776 (33.0) |
| FMM[a] (M, SD) |  | 0.18 (0.13) | 0.18 (0.14) | 0.17 (0.14) | 0.17 (0.13) | 0.17 (0.13) | 0.16 (0.14) | 0.16 (0.13) | 0.16 (0.13) | 0.15 (0.13) |
| LTCs[b] (n, %) | 0 | 507 (21.2) | 342 (16.8) | 270 (14.5) | 374 (16.8) | 345 (14.8) | 353 (14.2) | 317 (13.2) | 307 (12.9) | 285 (12.1) |
|  | 1 | 742 (31.0) | 588 (28.8) | 518 (27.9) | 589 (26.5) | 612 (26.3) | 630 (25.3) | 576 (24.0) | 542 (22.7) | 534 (22.7) |
|  | 2+ | 1147 (47.9) | 1109 (54.4) | 1068 (57.5) | 1260 (56.7) | 1369 (58.9) | 1506 (60.5) | 1512 (62.9) | 1534 (64.4) | 1534 (65.2) |

[a]FFM: Functional frailty measure;
[b]LTCs: Long-term conditions.

**Table 2. Summary statistics for five determinants and FMM for females across the nine waves.**

| | Wave | 1 | 2 | 3 | 4 | 5 | 6 | 7 | 8 | 9 |
|---|---|---|---|---|---|---|---|---|---|---|
| | N | 2965 | 2543 | 2333 | 2653 | 2769 | 2842 | 2836 | 2771 | 2828 |
| Age (M, SD) | | 74.12 (6.45) | 74.32 (6.43) | 74.63 (6.59) | 73.96 (6.46) | 74.14 (6.59) | 73.82 (6.52) | 73.88 (6.51) | 74.04 (6.53) | 74.21 (6.62) |
| Education (n, %) | High | 373 (12.6) | 372 (14.6) | 427 (18.3) | 536 (20.2) | 620 (22.4) | 596 (21.0) | 649 (22.9) | 667 (24.1) | 868 (30.7) |
| | Med or foreign | 822 (27.7) | 784 (30.8) | 800 (34.3) | 947 (35.7) | 1061 (38.3) | 1179 (41.5) | 1226 (43.2) | 1262 (45.5) | 1243 (44.0) |
| | low | 1770 (59.7) | 1387 (54.5) | 1106 (47.4) | 1170 (44.1) | 1088 (39.3) | 1067 (37.5) | 961 (33.9) | 842 (30.4) | 717 (25.4) |
| Net wealth (n, %) | Rich | 979 (33.0) | 829 (32.6) | 791 (33.9) | 888 (33.5) | 949 (34.3) | 1010 (35.5) | 1053 (37.1) | 1002 (36.2) | 1110 (39.3) |
| | Average | 577 (19.5) | 507 (19.9) | 474 (20.3) | 580 (21.9) | 574 (20.7) | 631 (22.2) | 616 (21.7) | 571 (20.6) | 605 (21.4) |
| | Poor | 1409 (47.5) | 1207 (47.5) | 1068 (45.8) | 1185 (44.7) | 1246 (45.0) | 1201 (42.3) | 1167 (41.1) | 1198 (43.2) | 1113 (39.4) |
| FMM[a] (M, SD) | | 0.22 (0.15) | 0.22 (0.15) | 0.21 (0.15) | 0.21 (0.14) | 0.21 (0.15) | 0.19 (0.15) | 0.19 (0.15) | 0.19 (0.14) | 0.18 (0.14) |
| LTCs[b] (n, %) | 0 | 500 (16.9) | 325 (12.8) | 249 (10.7) | 298 (11.2) | 282 (10.2) | 271 (9.5) | 278 (9.8) | 254 (9.2) | 261 (9.2) |
| | 1 | 881 (29.7) | 650 (25.6) | 542 (23.2) | 625 (23.6) | 588 (21.2) | 608 (21.4) | 568 (20.0) | 528 (19.1) | 532 (18.8) |
| | 2+ | 1584 (53.4) | 1568 (61.7) | 1542 (66.1) | 1730 (65.2) | 1899 (68.6) | 1963 (69.1) | 1990 (70.2) | 1989 (71.8) | 2035 (72.0) |

[a]FFM: Functional frailty measure;

[b]LTCs: Long-term conditions.

increased while the rate for those with low education decreased over time. Gradually, wealthy participants increased while the poorer participants declined. The healthy participants, who do not have any health conditions or have only one, declined over time, but those with two health conditions or more had increased over time.

The prevalence of LTCs in ELSA supports the definition of multimorbidity as two or more conditions. We observed that the count of two LTCs were steady across time for the participants, but it increased when participants had three or more LTCs (not shown here).

The number of male and female individuals varied over nine waves. For males 1271 (23%) were present in only one wave; two waves: 913 (16.5%); three waves: 795 (14.4%); four waves: 683 (12.4%); five waves: 533 (9.6%); six waves: 536 (9.7%); seven waves: 283 (5.1%); eight waves: 120 (4.5%) and nine waves 266 (4.8%). For females, 1369 (21.6%) were present in only one wave; two waves: 988 (15.6%); three waves: 897 (14.2%); four waves: 742 (11.7%); five waves: 637 (10.1%); six waves: 600 (9.5%); seven waves: 385 (6.1%); eight waves: 307 (4.9%) and nine waves 400 (6.3%). On average, the male and female individuals participated in 3.7 and 3.9 waves, respectively.

### Regression spline between FFM and age

Fig 2a shows the smoothed relationship between the FFM against age grouped by gender using regression splines. It shows that females have a higher frailty rate than males over time, and indicates that the relationship between age and FMM is non-linear. Also, the FFM increases as the number of LTCs increase for males, as shown in Fig 2b, and for females, as shown in Fig 2c. Moreover, multimorbid participants on average have a higher frailty as they aged in both genders.

### Multilevel growth model

Table 3 shows the outputs of the multilevel growth models for males and females. The general mean for the FFM for male is 0.17 (SD = 0.12) while for females is 0.20 (SD = 0.13) as shown in unadjusted models. AIC were improved for both models after adjusting to scale age, net wealth, education, two-way interaction between scale age, age$^2$ and LTCs categories.

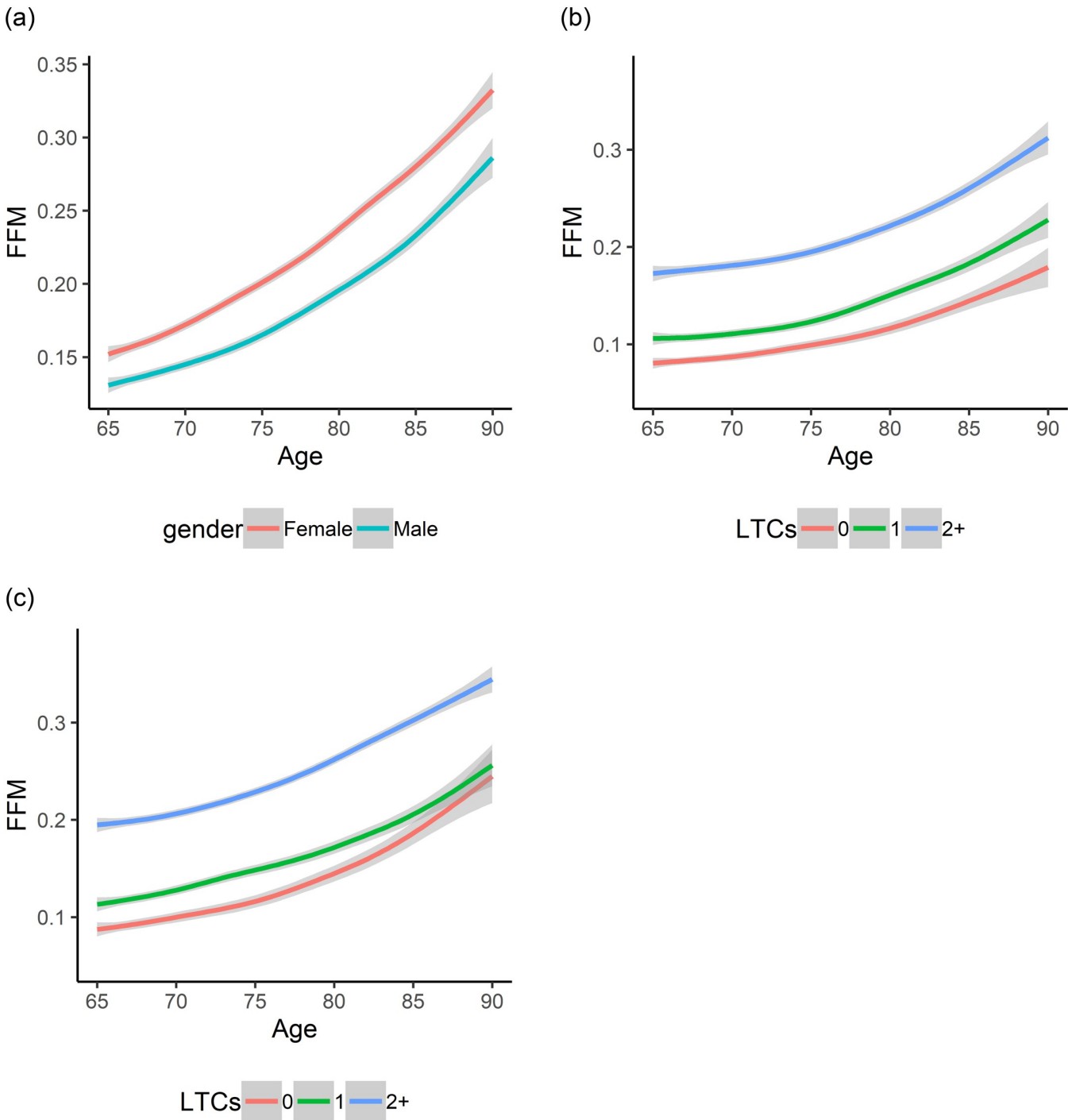

**Fig 2. FMM against age using regression splines.** a) FMM vs age grouped by gender. b) FMM vs age grouped by gender LTCs for males. c) FMM vs age grouped by gender LTCs for females. multimorbid participants on average have a higher frailty as they aged in both genders.

Considering the variation between the participants by adding age and age² as random effects contributed to better fit for male and female models.

In the adjusted models, frailty (as measured by FFM) increases nonlinearly for males with 4% each 10 years for the participants with no LTCs, average wealth and middle education,

**Table 3. Conditional multilevel growth model for FMM for male and female.**

| | Male | | | | Female | | | |
| --- | --- | --- | --- | --- | --- | --- | --- | --- |
| | Unadjusted model | | Adjusted model | | Unadjusted model | | Adjusted model | |
| **Fixed effects** | β | RSE | β | RSE | β | RSE | β | RSE |
| (Intercept) | 0.17* | 0.002 | 0.12* | 0.003 | 0.20 | 0.002 | 0.14* | 0.004 |
| Age c(75) /10 | | | 0.03* | 0.004 | | | 0.05* | 0.004 |
| Age$^2$ c(75) /10 | | | 0.01* | 0.005 | | | 0.01* | 0.005 |
| **Net wealth (ref. Average)** | | | | | | | | |
| Rich | | | -0.01* | 0.002 | | | -0.01* | 0.002 |
| Poor | | | 0.02* | 0.002 | | | 0.02* | 0.002 |
| **Education (ref. Middle)** | | | | | | | | |
| High | | | -0.01* | 0.003 | | | -0.01* | 0.003 |
| Low | | | 0.03* | 0.003 | | | 0.03* | 0.003 |
| **Health condition (ref. HC = 0)** | | | | | | | | |
| HC (1) | | | 0.03* | 0.003 | | | 0.02* | 0.004 |
| HC (2$^+$) | | | 0.06* | 0.004 | | | 0.06* | 0.004 |
| Age c(75) /10: HC (1) | | | 0.01* | 0.005 | | | -0.003 | 0.005 |
| Age$^2$ c(75) /10: HC (1) | | | 0.01* | 0.006 | | | 0.004 | 0.006 |
| Age c(75) /10: HC (2$^+$) | | | 0.03* | 0.005 | | | 0.008* | 0.005 |
| Age$^2$ c(75) /10: HC (2$^+$) | | | 0.03* | 0.006 | | | 0.019* | 0.006 |
| **Random effects** | | lower | upper | | lower | upper | | lower | upper |

| | Male Unadj | lower | upper | Male Adj | lower | upper | Female Unadj | lower | upper | Female Adj | lower | upper |
| --- | --- | --- | --- | --- | --- | --- | --- | --- | --- | --- | --- | --- |
| Intercept (SD) | 0.12 | 0.11 | 0.12 | 0.11 | 0.11 | 0.11 | 0.13 | 0.13 | 0.13 | 0.12 | 0.12 | 0.12 |
| Age c(75) /10 (SD) | | | | 0.09 | 0.08 | 0.09 | | | | 0.06 | 0.07 | 0.07 |
| Age$^2$ c(75) /10 (SD) | | | | 0.07 | 0.06 | 0.07 | | | | 0.06 | 0.05 | 0.06 |
| Error (SD) | 0.08 | 0.08 | 0.08 | 0.06 | 0.06 | 0.06 | 0.08 | 0.08 | 0.08 | 0.07 | 0.06 | 0.07 |

| **Model fit** | | | | | | | | |
| --- | --- | --- | --- | --- | --- | --- | --- | --- |
| AIC | -35189.48 | | -40730.02 | | -40439.67 | | -46306.36 | |

*p < .01.

while it increased 6% per decade in females. The rich and educated participants (males and females) were less frail while the poor and uneducated participants were frailer. For both genders, as the number of LTCs increases, the FFM score also increases. Also, the acceleration of FMM increases for males with one LTC or more; however in females the acceleration of FMM increases when they have two LTCs or more. Fig 3 shows the acceleration of FMM for both genders are non-linear.

The adjusted multilevel growth models for complete cases showed similar results as the primary analysis, but the most two-way interaction terms in the male and female models were no longer significant (see Table 7 in the S1 Text).

## Discussion

In this study, we have shown that FFM increases with the number of LTCs, respectively, for males and females. We also found that the number of LTCs affects FFM progression for males with one LTC and more, and for females with two or more LTCs. Moreover, the relationship between age and FFM, the measure of frailty without LTCs, was nonlinear for those over 65.

Our findings were consistent with previous studies. Thompson, Theou [8] reported that multimorbid people have a higher frailty score than non-multimorbid people using the accumulation deficit model. Also, the effect of age and wealth status on frailty was consistent with

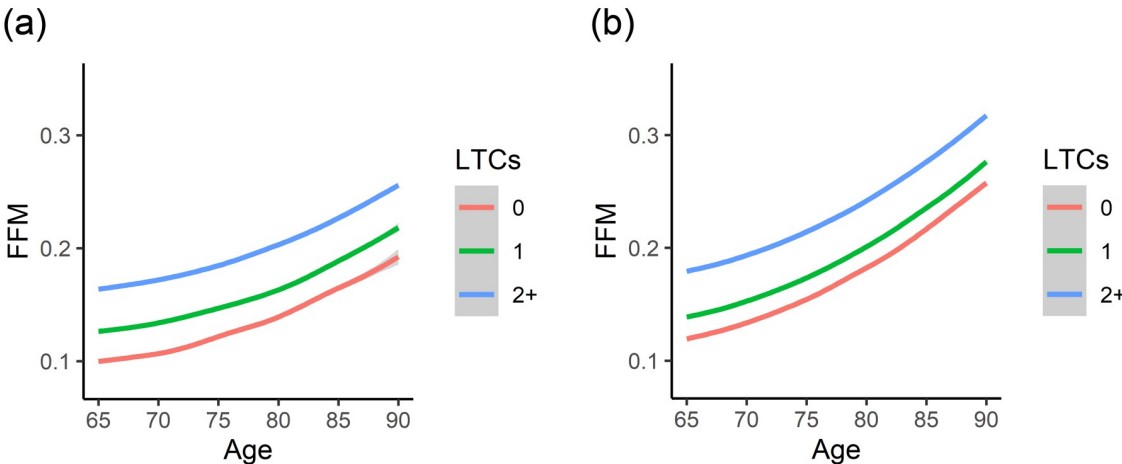

**Fig 3. The interaction between the age and FFM.** a): The interaction between the age and FFM grouped by LTCs for males. b): The interaction between the age and FFM grouped by LTCs for females. The acceleration of FMM increases for males with one LTC or more; however in females the acceleration of FMM increases when they have two LTCs or more.

the findings of Marshall, Nazroo [15] findings. They found that the younger and wealthy participants were less frail over time. Moreover, highly educated people are often less frail [28].

The number of wealthy and educated participants have increased over the nine waves, at the same time the participants with two or more LTCs increased, and the average of FFM scores across the time decreased. It can be evidence that a good wealth status and a high education contribute to delaying frailty progression or at least those wealthier and more educated participants can manage their health despite LTCs more efficiently than others.

Two LTCs could be a reasonable threshold for the health providers to commit more attention to dealing with those participants as soon as possible to delay frailty progression. Some LTC cases in males and females have to be considered separately with further care once the participant has Parkinson's or Dementia for males and Psychiatric and Alzheimer's disease for females.

It is preferred to apply the cumulative deficit model rather than a phenotype model to study the impact of frailty in order to create interventions that reduce frailty among older people [29]. For example, some participants might not benefit from a physical intervention program, such as a group exercise, due to their struggle with cognitive or psychological issues. Using a multidimensional frailty measure could be more appropriate to measure various frailty components for older people in a community, enabling identification of the weaker dimension(s) that can be targeted with intervention from health givers.

One of the strengths of this study is that it uses large, high-quality data from the English population with a large sample size. Furthermore, we were able to investigate the association of several factors with FFM development for an extended period (18 years) because of the vast collection of longitudinal data and large sample size available in the ELSA. Moreover, it supports studies that investigate the relationship between frailty and multimorbidity to do more research.

Study results should also be interpreted in light of potential limitations. We used the FFM rather than the frailty index, but we explained how the FMM is sufficient. Secondly, in the cumulative deficit model It is common to use "deficit" that are difficult to change and may not accurately reflect frailty's "reversible" nature. Also, researchers choose "deficit" items based on their subjectivity or what is available in the use dataset in the cumulative deficit model.

Nevertheless, this model is useful for within-cohort comparisons [30]. Next, the rate of missing data in some frailty deficits was relatively high. However, we applied two single imputation methods and then conducted the sensitivity analysis. Implementing multilevel models in longitudinal data might be more complicated because an individual's current measure may often be correlated to prior measures [24]. We observed the presence of autocorrelation among individual errors using the autocorrelation function (ACF). We treated this issue by applying robust standard error. Besides that, we should have included further determinants, such as physical activity and lifestyle determinants. However, many of them were measured only at wave one, which we cannot include in our analysis because it will increase the rate of missing data since the refreshment samples were not asked to report earlier determinants.

In summary, these results provide the first step to studying the effect of multimorbidity on frailty progression in older people who live in the England community. Health providers should be aware of planning a suitable intervention once the elderly have two or more health conditions. There is a further need to investigate the interactions between health conditions and frailty progression.

## Supporting information

**S1 Text. Supplementary file.**
(DOCX)

## Author Contributions

**Conceptualization:** Ali Alattas, Robert West.

**Data curation:** Ali Alattas.

**Formal analysis:** Ali Alattas, Farag Shuweihdi.

**Investigation:** Ali Alattas, Silviya Nikolova, Farag Shuweihdi, Robert West.

**Methodology:** Ali Alattas, Farag Shuweihdi, Robert West.

**Project administration:** Robert West.

**Software:** Ali Alattas, Farag Shuweihdi.

**Supervision:** Silviya Nikolova, Farag Shuweihdi, Kate Best, Robert West.

**Visualization:** Ali Alattas.

**Writing – original draft:** Ali Alattas.

**Writing – review & editing:** Silviya Nikolova, Kate Best, Robert West.

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
