## [Decision Letter · Decision Letter 0]

9 Mar 2023

PONE-D-23-03895The impact of long-term conditions on the progression of frailtyPLOS ONE

Dear Dr. Alattas,

Thank you for submitting your manuscript to PLOS ONE. After careful consideration, we feel that it has merit but does not fully meet PLOS ONE’s publication criteria as it currently stands. Therefore, we invite you to submit a revised version of the manuscript that addresses the points raised during the review process.

We look forward to receiving your revised manuscript.

Kind regards,

Mario Ulises Pérez-Zepeda, M.D., Ph.D.

Academic Editor

PLOS ONE

“This work was funded by the Cultural Bureau, Embassy of Saudi Arabia, London.  “

Reviewers' comments:

Reviewer's Responses to Questions

**Comments to the Author**

1. Is the manuscript technically sound, and do the data support the conclusions?

Reviewer #1: Yes

Reviewer #2: Partly

2. Has the statistical analysis been performed appropriately and rigorously? 

Reviewer #1: Yes

Reviewer #2: Yes

3. Have the authors made all data underlying the findings in their manuscript fully available?

Reviewer #1: Yes

Reviewer #2: Yes

4. Is the manuscript presented in an intelligible fashion and written in standard English?

Reviewer #1: Yes

Reviewer #2: Yes

5. Review Comments to the Author

Reviewer #1: The authors examined the association between the number of LTCs and functional frailty measurement (FFM) in a large longitudinal study of the general population in England. They found that as the number of long-term conditions (LTCs) increased, the degree of FFM also progressed and that while more than one LTC affected FFM in men, more than two LTCs involved FFM in women. This phenomenon is interesting. However, this paper raises the following issues.

The reviewer recommends describing the definition of LTC and the clinical conditions that LTC represents in the introduction part.

The authors define frailty using the cumulative deficit model. There are descriptions that the cumulative deficit model is preferred to the phenotypic model in the current study settings. The reviewer only agrees with this idea. The phenotypic model and the cumulative deficit model both have their advantages and disadvantages. In the cumulative deficit model, the choice of "deficit" items to add varies from study to study and may vary depending on the researcher's subjectivity. In addition, the components of the cumulative deficit model often use items that are difficult to change and may not reflect the "reversible" nature of frailty. Therefore, the reviewer recommends describing the limitations of the cumulative deficit model.

Reviewer #2: In this manuscript, the authors presented a correlation between the effects of multimorbidity on Frailty progression in older adults in the English population. With the advancement of different statistical methods coupled with data analysis tools, it is highly imperative that we start correlating findings related to patients’ disease progression and plan counter-exercise therapy tools, especially for older adults with multi symptoms. The study here is a solid attempt in that direction. Precisely, incorporating data from a large sample size for both females and males across nine different waves was impressive and significant enough to draw a rational conclusion in an otherwise debatable topic of the longitudinal impact of multiple long-term conditions (LTCs) on Frailty progression.

The authors used the functional frailty measurement (FFM) technique for determining frailty progression and found that the progression accelerates differently in the female and males population. The FFM increased 4% every 10 years for the male participants with no LTCs, while it increased 6% per decade in females. At the same time, the acceleration of FMM increases for males with one long-term health condition or more; however, in females, the acceleration of FMM increases when they have two LTCs or more.

This is a well-correlated and conducted study with robust statistics and analytical methods. The FFM approach taken was well justified by the author with relevant references. The incorporation of secondary determinants like age, education level, and net worth was really nice and added a lot of value and gives a solid perspective to the overall outcome of the study.

Additionally, using two different approaches for handling missing data sets is highly appreciative and in fact, it reduced the chances of misinterpretation of the data.

While there is always scope for further investigation and incorporation of many other factors in this study, however at this point this study stands out to be a much more data-driven conclusion and presumptive especially with the involvement of so many factors which are highly relative.

The only caveat I see in this study is the figure representation, especially the line graphs (figures 2 and 3). It is recommended to remove background lines from each figure, adding solid X and Y-axis lines, and giving the entire figure a much more professional publishable look. The same is applicable to supplementary figures as well. Secondly, there are no figure legends and this needs to be added.

6. PLOS authors have the option to publish the peer review history of their article (what does this mean?). If published, this will include your full peer review and any attached files.

Reviewer #1: No

Reviewer #2: No

---

## [Author Response · Author response to Decision Letter 0]

14 Mar 2023

Reviewer 1: We added two limitations to the cumulative deficit model in the discussion section and defined long-term conditions (LTCs) in the introduction. However, the 16 clinical conditions were listed in the method section, and we believe it is appropriate to record them there. 

Reviewer 2: We changed the background of the figures in the manuscript and added the legends, but we did not do that for the supplementary file.

---

## [Editor Report · Decision Letter 1]

23 Mar 2023

The impact of long-term conditions on the progression of frailty

PONE-D-23-03895R1

Dear Dr. Alattas,

We’re pleased to inform you that your manuscript has been judged scientifically suitable for publication and will be formally accepted for publication once it meets all outstanding technical requirements.

Kind regards,

Mario Ulises Pérez-Zepeda, M.D., Ph.D.

Academic Editor

PLOS ONE

---

## [Editor Report · Acceptance letter]

28 Mar 2023

PONE-D-23-03895R1 

The impact of long-term conditions on the progression of frailty 

Dear Dr. Alattas:

I'm pleased to inform you that your manuscript has been deemed suitable for publication in PLOS ONE. Congratulations! Your manuscript is now with our production department. 

Kind regards, 

on behalf of

Dr. Mario Ulises Pérez-Zepeda 

Academic Editor

PLOS ONE